# Learning ECG Representations via Poly-Window Contrastive Learning

Yi Yuan*[†], Joseph Van Duyn*[†], Runze Yan [‡], Zhuoyi Huang[¶], Sulaiman Vesal [¶], Sergey Plis[§],
Xiao Hu[‡], Gloria Hyunjung Kwak[‡], Ran Xiao[‡], Alex Fedorov[‡]
*Department of Quantitative Theory & Methods
[†]Department of Mathematics
[‡]Center for Data Science, Nell Hodgson Woodruff School of Nursing
Emory University, Atlanta, GA, USA
[§] Department of Computer Science, Georgia State University, Atlanta, GA, USA
[¶] Microsoft

*Abstract*—**Electrocardiogram (ECG) analysis is foundational for cardiovascular disease diagnosis, yet the performance of deep learning models is often constrained by limited access to annotated data. Self-supervised contrastive learning has emerged as a powerful approach for learning robust ECG representations from unlabeled signals. However, most existing methods generate only pairwise augmented views and fail to leverage the rich temporal structure of ECG recordings. In this work, we present a poly-window contrastive learning framework. We extract multiple temporal windows from each ECG instance to construct positive pairs and maximize their agreement via statistics. Inspired by the principle of slow feature analysis, our approach explicitly encourages the model to learn temporally invariant and physiologically meaningful features that persist across time. We validate our approach through extensive experiments and ablation studies on the PTB-XL dataset. Our results demonstrate that poly-window contrastive learning consistently outperforms conventional two-view methods in multi-label superclass classification, achieving higher AUROC (0.891 vs. 0.888) and F1 scores (0.680 vs. 0.679) while requiring up to four times fewer pre-training epochs (32 vs. 128) and 14.8% in total wall clock pre-training time reduction. Despite processing multiple windows per sample, we achieve a significant reduction in the number of training epochs and total computation time, making our method practical for training foundational models. Through extensive ablations, we identify optimal design choices and demonstrate robustness across various hyperparameters. These findings establish poly-window contrastive learning as a highly efficient and scalable paradigm for automated ECG analysis and provide a promising general framework for self-supervised representation learning in biomedical time-series data.**

*Index Terms*—**Electrocardiogram, Self-supervised learning, Contrastive learning, Deep Learning**

## I. Introduction

Cardiovascular disease (CVD) remains the leading cause of death globally. Electrocardiogram (ECG) analysis plays a critical role in modern clinical care, as ECG is a widely accessible, non-invasive, and cost-effective tool for detecting cardiac abnormalities and CVD risk stratification [1]. However, accurate ECG interpretation is time-consuming and requires specialized expertise and is subject to human error [2]. Recent advances in deep learning (DL) have significantly improved ECG analysis to detect subtle pathologies and facilitate earlier diagnosis [3]–[5]. Nevertheless, the reliance of DL models on large labeled datasets limits their practical deployment in real-world medical settings since high-quality expert annotations are scarce and expensive to obtain [6].

Self-supervised learning (SSL) has emerged as a powerful paradigm for representation learning in the absence of manual labels [7]–[11]. In particular, contrastive learning methods such as SimCLR [10], MoCo [12], and their adaptations to time-series data [6], [13] have demonstrated strong performance by maximizing agreement between different augmented views of the same instance, thus learning robust and generalizable features from unlabeled data.

Most contrastive learning frameworks in the biomedical time series analysis [13]–[16] generate only two augmented views for each input. This common approach does not make full use of the rich temporal structure found in ECG signals. In clinical practice, a typical 10-second ECG recording captures several cardiac cycles, and the rhythm and shape of these cycles are usually similar throughout the recording. Pathological features, if they exist, also tend to be present across the entire signal. Therefore, both normal and abnormal heart dynamics can be observed in multiple, separate time windows within one ECG. By focusing only on pairwise comparisons, traditional contrastive methods do not take advantage of this persistent physiological structure. Each segment of the ECG is produced by the underlying electrical activity of the heart. As a result, we expect that a model that uses information from multiple segments can learn features that better reflect the real physiological sources.

Recent theoretical work [17] has shown that contrastive learning can be improved by using more than two positive views for each sample. By using multiple views, these methods can capture more of the variability within each sample, leading to faster and more efficient learning. Poly-view contrastive learning can improve the learned representations for natural images, but so far, it has not been widely applied to physiological time-series data like ECG signals.

Corresponding author: avfedor@emory.edu

The temporal patterns in cardiac signals also offer an opportunity to learn features that are stable over time. This idea is related to the concept of *slow feature analysis* (SFA) [18], which aims to find features that change slowly across time and thus capture the main structure in time series data. For ECG analysis, this is important because heart rhythms and arrhythmias usually last over several cycles, while noise and artifacts are transient and inconsistent.

Inspired by SFA [18], [19] and poly-view contrastive learning [17], we propose a **poly-window contrastive learning** framework for ECG representation learning. We extract multiple, temporally diverse windows from each ECG and maximize similarity across these pairs via arithmetic or geometric mean. We hypothesize that it allows the model to capture slow, physiologically relevant features that persist across the ECG recording. Unlike conventional contrastive approaches limited to pairwise views, our strategy with multiple windows directly exploits the temporal redundancy and structure of clinical ECGs.

We validate our approach on the large-scale PTB-XL dataset [20], demonstrating consistent improvements over strong baselines in multi-label superclass classification. Through extensive ablation and statistical analysis, we show that poly-window contrastive learning enables rapid convergence, improves linear evaluation performance, and provides robustness to hyperparameter selection. Our findings support the ideas of slow feature analysis principles in poly-view contrastive objectives. We achieve the state-of-the-art performance on PTB-XL for the contrastive approach and significantly accelerate training by 14.8%, enhancing sample efficiency.

## II. METHODS

### A. Poly-Window Contrastive Learning

We propose a Poly-Window Contrastive Learning objective [17] that extends the standard two-view contrastive framework by extracting $M$ temporal windows from each ECG recording. Instead of relying on a single positive pair, we compute similarity across all window pairs and aggregate them via a statistic (e.g., arithmetic or geometric mean). This encourages the encoder to learn features that are stable across both local distortions and longer temporal shifts.

In ECGs, rhythms and morphological patterns typically persist across cardiac cycles while noise and artifacts are transient. Enforcing similarity across multiple windows leverages this physiological redundancy. Our approach is further motivated by principles of slow feature analysis (SFA) [18], which aim to extract features that change slowly over time. By enforcing consistency among multiple windows, the model captures stable cardiac dynamics rather than overfitting to local fluctuations. This poly-window strategy directly exploits the temporal redundancy of ECGs and yields temporally invariant, physiologically meaningful representations.

Our framework is illustrated in Fig. 1. For each ECG in a batch of size $N$, we sample $M$ (possibly overlapping) temporal windows, resulting in $NM$ windows. Each window is independently encoded, yielding a matrix of L2-normalized embeddings $\mathbf{Z} \in \mathbb{R}^{(NM) \times d}$, where $d$ is the embedding dimension. We compute a pairwise similarity matrix:

$$\mathbf{S} = \frac{1}{\tau} \mathbf{Z} \mathbf{Z}^\top, \tag{1}$$

where $\tau$ is a temperature parameter and $S_{ab}$ is the scaled cosine similarity between representations $z_a$ and $z_b$.

Let $M_{\text{pos}} \in \{0,1\}^{(NM) \times (NM)}$ be a binary mask, where $M_{\text{pos}}(a,b) = 1$ if $z_a$ and $z_b$ are different windows of the same signal, and $0$ otherwise. For each anchor $z_a$, define $\mathcal{P}(a)$ as the set of indices $b$ for which $M_{\text{pos}}(a,b) = 1$ (i.e., all other windows of the same sample as $z_a$). The self-similarity mask is defined by $a = b$.

To generalize beyond the standard two-view contrastive loss, we introduce a poly-window formulation that aggregates all positive pairs for each anchor using a chosen statistic (e.g., arithmetic or geometric mean). The choice of aggregation reflects different assumptions: the arithmetic mean assumes additive contributions but can overweight outlier pairs, while the geometric mean is less sensitive to outliers. Other sufficient statistics could be considered in the latent space [17], but it has not been shown to improve performance.

The resulting batch-averaged poly-window InfoNCE [7] contrastive losses are:

$$\mathcal{L}_{\text{geo}} = -\frac{1}{NM} \sum_{a=1}^{NM} \Big[ \frac{1}{M-1} \sum_{b \in \mathcal{P}(a)} S_{ab} - \log\Big( \sum_{c \neq a} e^{S_{ac}} \Big) \Big], \tag{2}$$

$$\mathcal{L}_{\text{arith}} = \frac{1}{NM} \sum_{a=1}^{NM} \Big[ \log(M-1) + \log\Big( \sum_{c \neq a} e^{S_{ac}} \Big) - \log\Big( \sum_{b \in \mathcal{P}(a)} e^{S_{ab}} \Big) \Big]. \tag{3}$$

For the special case $M = 2$, each anchor $z_a$ has a single positive $z_{a^+}$, and $M - 1 = 1$, so both forms reduce to the standard InfoNCE loss [7]:

$$\mathcal{L}_{\text{geo}} = -\frac{1}{2N} \sum_{a=1}^{2N} \Big[ S_{a,a^+} - \log\Big( \sum_{c \neq a} e^{S_{ac}} \Big) \Big]. \tag{4}$$

### B. Derivation of loss functions

We now provide the detailed derivation. We show how the aggregation statistic is introduced into the InfoNCE. The InfoNCE with Geometric mean aggregation is defined as follows:

$$\mathcal{L}_{\text{geo}} = -\frac{1}{NM} \sum_{a=1}^{NM} \log\Big( \Big[ \prod_{b \in \mathcal{P}(a)} p_{ab} \Big]^{1/(M-1)} \Big)$$

$$= -\frac{1}{NM} \sum_{a=1}^{NM} \frac{1}{M-1} \sum_{b \in \mathcal{P}(a)} \log p_{ab}$$

$$= -\frac{1}{NM} \sum_{a=1}^{NM} \frac{1}{M-1} \sum_{b \in \mathcal{P}(a)} \Big[ S_{ab} - \log\Big( \sum_{c \neq a} e^{S_{ac}} \Big) \Big]$$

$$= -\frac{1}{NM} \sum_{a=1}^{NM} \Big[ \frac{1}{M-1} \sum_{b \in \mathcal{P}(a)} S_{ab} - \log\Big( \sum_{c \neq a} e^{S_{ac}} \Big) \Big],$$

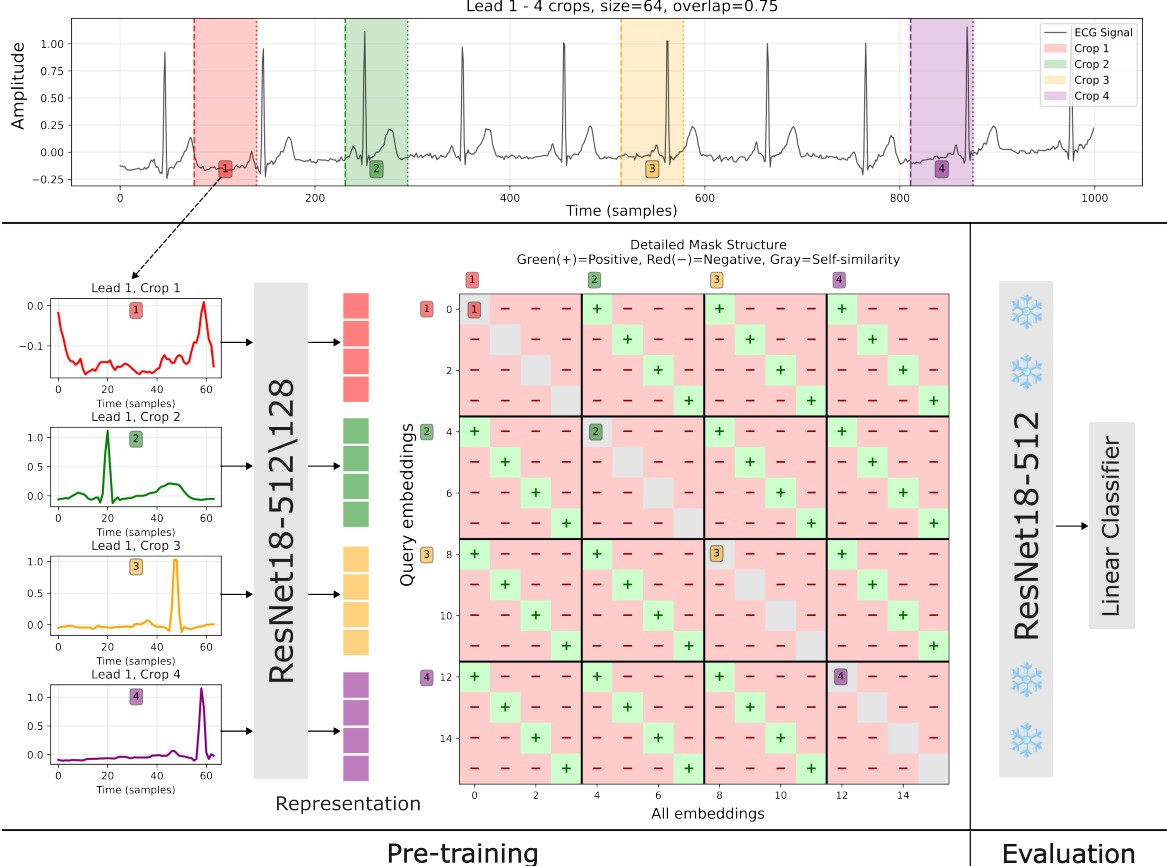

Fig. 1: **Poly-Window Contrastive Learning for ECG Signals.** (**Top**) Simplified visualization with one single lead from a 12-lead ECG recording (1000 time-points). In our framework, the temporal crops are applied across all leads, and all leads are used as one input. (**Left**) Waveforms for each colored crop, which are encoded by a ResNet-18 backbone into 512-dimensional embeddings (projected to 128 dimensions for contrastive learning). (**Right**) Poly-window contrastive similarity matrix: green "+" entries denote positive pairs (different windows from the same recording), red "-" entries denote negative pairs (windows from different recordings), and gray diagonal entries (self-similarities) are excluded from the loss. Note that the positive pairs marked here are also used to compute the denominator in the contrastive objective. During evaluation, the pre-trained encoder is frozen and a linear classifier is trained on the learned representations.

where $p_{ab} = \frac{e^{S_{ab}}}{\sum_{c \neq a} e^{S_{ac}}}$.

The InfoNCE with the Arithmetic mean aggregation is defined as follows:

$$\mathcal{L}_{\text{arith}} = -\frac{1}{NM} \sum_{a=1}^{NM} \log\left(\frac{1}{M-1} \sum_{b \in \mathcal{P}(a)} p_{ab}\right)$$

$$= -\frac{1}{NM} \sum_{a=1}^{NM} \log\left(\frac{1}{M-1} \sum_{b \in \mathcal{P}(a)} \frac{e^{S_{ab}}}{\sum_{c \neq a} e^{S_{ac}}}\right)$$

$$= -\frac{1}{NM} \sum_{a=1}^{NM} \log\left(\frac{1}{M-1} \cdot \frac{1}{\sum_{c \neq a} e^{S_{ac}}} \sum_{b \in \mathcal{P}(a)} e^{S_{ab}}\right)$$

$$= -\frac{1}{NM} \sum_{a=1}^{NM} \left[-\log(M-1) - \log \sum_{c \neq a} e^{S_{ac}} + \log\left(\sum_{b \in \mathcal{P}(a)} e^{S_{ab}}\right)\right]$$

$$= \frac{1}{NM} \sum_{a=1}^{NM} \left[\log(M-1) + \log\left(\sum_{c \neq a} e^{S_{ac}}\right) - \log\left(\sum_{b \in \mathcal{P}(a)} e^{S_{ab}}\right)\right].$$

Our poly-window loss aggregates all $M-1$ positives per anchor via statistics, rather than evaluating every positive pair independently. This reduces the positive-side accounting from $O(M^2 N^2)$ to $O(MN)$ by computing a single $MN \times MN$ similarity matrix and summarizing each anchor's positives to $MN$ summaries. We hypothesize that it improves the signal-to-noise ratio of the gradient update, consistent with the reduced estimator variance and tighter mutual information bounds observed for poly-view objectives with higher view multiplicity [17].

### C. Dataset

For all experiments, we used the PTB-XL dataset [20], a large-scale, publicly available collection of $21,837$ clinical 12-lead ECG recordings (10 seconds each, sampled at 100Hz) across a wide age range and balanced by sex. All models were trained and evaluated using the complete set of 12 leads as one input for each ECG recording. Each ECG was annotated by

TABLE I: Hold-out test linear evaluation performance on Multi-label Superclass classification as mean [95% CI].

| Windows | Batch Size | Loss Type | Crop | Overlap | Epochs ↓ | Loss Func | F1 Score ↑ | AUROC ↑ | Recall ↑ | Precision ↑ |
|---|---|---|---|---|---|---|---|---|---|---|
| 2 | 256 | Geometric | 64 | 0.00 | 128 | Geometric | 0.679 [0.667, 0.690] | 0.888 [0.884, 0.893] | **0.660** [0.640, 0.680] | **0.710** [0.685, 0.735] |
| 4 | 768 | Geometric | 64 | 0.50 | 64 | Geometric | 0.675 [0.666, 0.685] | 0.887 [0.886, 0.888] | 0.665 [0.645, 0.684] | 0.700 [0.689, 0.710] |
| 6 | 256 | Geometric | 64 | 0.75 | **32** | Geometric | 0.675 [0.669, 0.680] | 0.890 [0.887, 0.892] | 0.656 [0.637, 0.676] | 0.704 [0.685, 0.723] |
| 8 | 256 | Geometric | 64 | 0.50 | **32** | Geometric | **0.680** [0.672, 0.688] | **0.891** [0.889, 0.893] | **0.660** [0.642, 0.678] | **0.710** [0.699, 0.722] |

up to two cardiologists following the SCP-ECG standard, with diagnostic labels organized hierarchically into superclasses, subclasses, and fine-grained multi-label statements covering all 71 classes. We conducted classification experiments at the superclass granularity level. The five superclasses used for multi-label classification in PTB-XL are: Normal ECG (NORM), Myocardial Infarction (MI), ST/T Change (STTC), Conduction Disturbance (CD), and Hypertrophy (HYP). Following the dataset's recommended evaluation protocol, we used the standard stratified splits: folds 1–8 for training, fold 9 for validation, and fold 10 as the held-out test set. To ensure robustness, all results were averaged over five runs with different random seeds.

### D. Experimental setup

Our experimental workflow consists of two main stages: the first stage is self-supervised pre-training, and the second stage consists of linear evaluation.

We begin by pre-training a 1D variant of ResNet 18 on the PTB-XL dataset using a contrastive learning approach. The ResNet18 model encodes 12-lead ECG temporal windows and projects the time-series into 512-dimensional representations. Then we embedded this representation into 128-dimensional space via a 1-layer linear projection head to compute similarity for the contrastive objective. We note such architecture as ResNet18-512/128. During pre-training, the input size is defined by specifying a window crop size (32, 64, 128, 256 time points), while during linear evaluation, we use all the 1000 time points. We do not apply any other augmentation, only random stratified temporal cropping with an allowable overlap. For pre-training, models were trained for 500 epochs using the PyTorch [21] framework with mixed-precision training. The AdamW [22] optimizer was employed with an initial learning rate of 0.01, weight decay of $1 \times 10^{-4}$, and $\epsilon$ of $1 \times 10^{-8}$. The learning rate was scheduled using a linear warmup for the first 10 steps, followed by cosine decay to a final learning rate of $1 \times 10^{-6}$ over the full training schedule.

After pre-training, we take the model weights from the last checkpoint (final epoch). We remove 128-dimensional projection head, and freeze the encoder up to the last adaptive average pooling layer, which returns 512-dimensional representation (we note it as ResNet18-512). Then we train a linear classifier to assess the quality of learned 512-dimensional features. For linear evaluation, we train a classifier for 90 epochs using the same optimizer and learning rate scheduler, and we optimize a multi-label one-versus-all classification objective using PyTorch's `MultiLabelSoftMarginLoss`. The model selection is based on the best validation performance with the F1 metric.

## III. RESULTS

In this section, we present the empirical evaluation and ablations of our proposed poly-window contrastive learning framework on the multi-label superclass classification task.

### A. Linear evaluation performance on superclass classification

Table I summarizes the hold-out test set results for linear evaluation on the multi-label superclass classification task. All values are reported as the mean with 95% confidence intervals, calculated over multiple independent runs. The metrics evaluated include F1 score, AUROC, recall, and precision.

Across all configurations, we observe robust and consistent model performance. The highest AUROC is achieved by the configuration with 8 windows, batch size 256, crop size 64, overlap 0.50, and 32 training epochs (0.891 [0.889, 0.893]) with the best F1 score (0.680 [0.672, 0.688]), recall (0.660 [0.642, 0.678]), and precision (0.710 [0.699, 0.722]). However, pairwise comparisons between the top-performing methods, performed using the Wilcoxon signed-rank test with Holm's correction ($p > 0.05$), revealed no statistically significant differences in performance.

A notable finding is that increasing the number of windows allows the model to reach optimal or near-optimal performance at a substantially lower number of training epochs. Specifically, for 6 and 8 windows, saturation is observed at only 32 epochs. At the same time, configurations with fewer windows require more epochs to converge. This observation shows the advantage of the poly-window strategy, as higher view multiplicity accelerates convergence without sacrificing performance.

### B. Ablations

We conducted a series of ablation studies to systematically assess the contribution of each hyperparameter. Each ablation isolates the effect of a single factor by varying one parameter at a time. All results are reported on the validation set.

*1) Effect of crop size:* The influence of crop size on the poly-window contrastive learning framework is depicted in Figure 2a. The model achieves the highest AUROC at a crop size of 64. Surprisingly, this performance occurs when the temporal window captures approximately one cardiac cycle. It may suggest that our representations capture the dynamics of these physiological processes within one beat. Both larger crops (128, 256) and very short crops (32) yield lower performance, likely due to either capturing multiple beats or lacking sufficient temporal context, respectively. Crop size 64 outperforms all other tested crop sizes ($p < 0.00001$), with a large effect size ($\eta^2 = 0.5491$), highlighting crop size as the most critical hyperparameter.

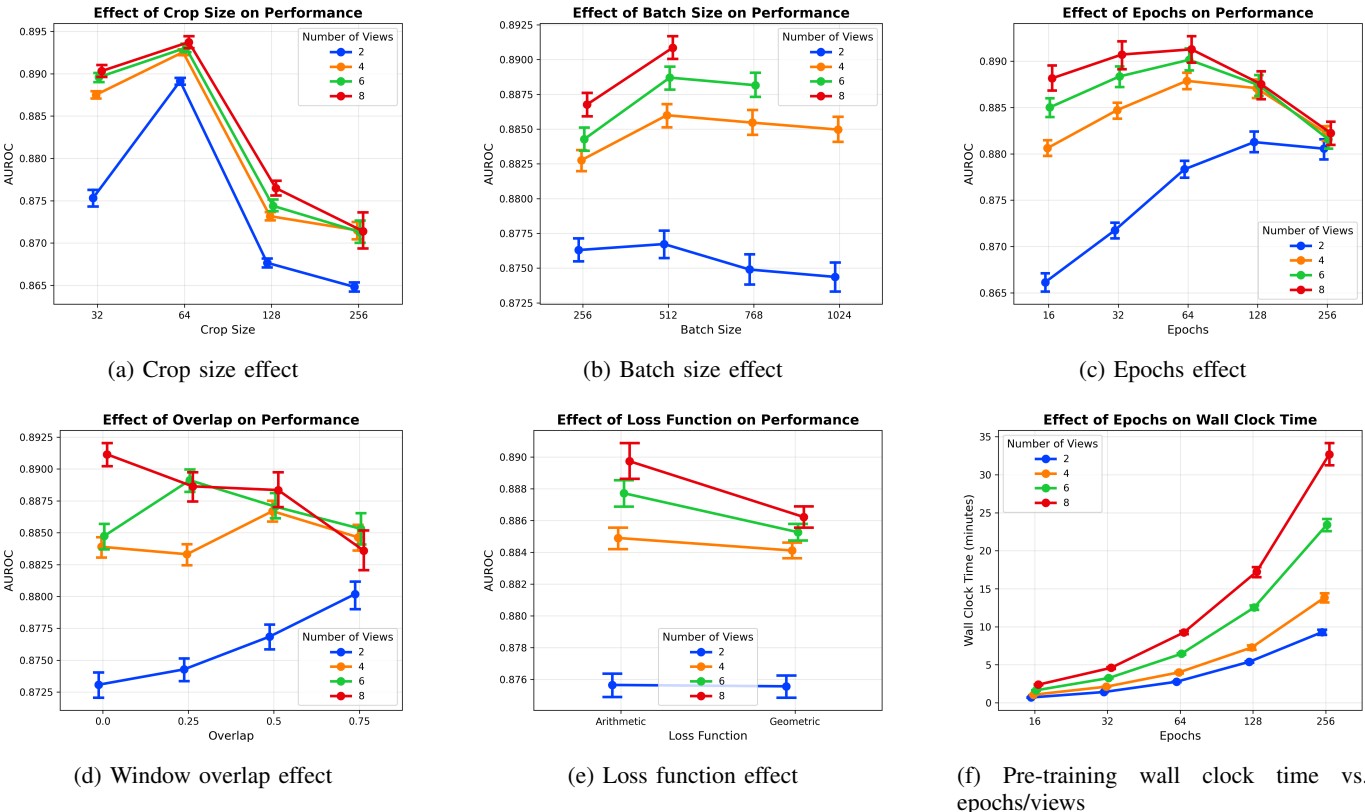

(a) Crop size effect

(b) Batch size effect

(c) Epochs effect

(d) Window overlap effect

(e) Loss function effect

(f) Pre-training wall clock time vs. epochs/views

Fig. 2: **Ablation and wall-clock time results for the poly-window contrastive learning framework** Validation AUROC and wall clock time are shown across an extensive grid search covering: (NUM_VIEWS = 2, 4, 6, 8), (CROP_SIZES = 32, 64, 128, 256), (OVERLAPS = 0.0, 0.25, 0.5, 0.75), (NUM_EPOCHS = 16, 32, 64, 128, 256), (SEEDS = 0, 42, 123, 555, 789), loss functions (LOSS_FUNCS = geometric, arithmetic), and batch sizes (BATCH_SIZES = 256, 512, 768, 1024). Each subplot reports the mean and CI95 across five random seeds. (a) Effect of crop size on model performance (AUROC). (b) Effect of batch size on AUROC. (c) Effect of number of training epochs on AUROC. (d) Effect of window overlap on AUROC. (e) Effect of loss function on AUROC. (f) Effect of training epochs and number of views on wall clock time.

*2) Effect of batch size:* As shown in Figure 2b, AUROC improves as the batch size increases from 256 to 512. However, further increasing the batch size to 768 and 1024 results in a decrease in performance, likely due to diminishing returns and less effective gradient estimates at larger batch sizes. The effect size for batch size is small ($\eta^2 = 0.0144$, $p < 0.00001$). All experiments utilized the AdamW optimizer. The potential influence of alternative optimization algorithms remains an open question for future research.

*3) Effect of number of epochs:* Figure 2c presents model performance as a function of training epochs. While AUROC generally increases with additional epochs, the optimal number of epochs is strongly influenced by the number of windows. For a number of windows of two, the best AUROC performance is achieved with 128 epochs. When the number of windows exceeds 4, optimal performance is reached much earlier, typically within 32 to 64 epochs. This demonstrates that leveraging a higher number of windows enables the model to converge more rapidly. The observed Effect of epochs is small ($\eta^2 = 0.0574$, $p < 0.00001$).

*4) Effect of overlap:* Figure 2d summarizes the effect of window overlap on model performance. The observed effect

size is negligible ($\eta^2 = 0.0063$, $p < 0.00001$), hence the practical impact of overlap is limited. However, for two windows, it can improve performance slightly, while for more windows, the performance might decrease.

*5) Effect of loss function:* As shown in Figure 2e, the arithmetic loss function consistently achieves a higher mean AUROC compared to the geometric loss across all numbers of windows. This pattern suggests that arithmetic aggregation is marginally more effective for maintaining the discriminative structure of the learned representations. However, the difference between loss functions is quantitatively small with negligible effect size ($\eta^2 = 0.0075$, $p < 0.00001$).

*6) Details and on statistical analysis and interactions.:* We performed over $7,200$ experiments to validate the effects for crop size, overlap, epochs, loss function, number of windows, and batch size. Among these, crop size ($\eta^2 = 0.5491$) and number of windows ($\eta^2 = 0.1526$) exhibit large effect sizes and are the most influential hyperparameters. Interaction analyses identify a medium effect for the crop size × number of window interactions ($\eta_p^2 = 0.0648$).

TABLE II: Comparison with previous work on **superclass multi-label classification** (F1 and AUROC). All methods were trained and evaluated on PTB-XL unless otherwise noted. "Architecture" indicates the model backbone and embedding dimension (e.g., ResNet50/2048). "Epochs" are reported as pre-training/linear evaluation. The "repr." are noted as reproduced approaches by a specific citation. N/R means that the metrics are not reported.

| Method | Architecture | Epochs | F1 Score | AUROC |
|---|---|---|---|---|
| SimCLR repr. by [13] | ResNet18/256 | 400/100 | N/R | 0.648 |
| CLOCS repr. by [13] | ResNet18/256 | 400/100 | N/R | 0.784 |
| 3KG repr. by [13] | ResNet18/256 | 400/100 | N/R | 0.751 |
| TSSL [13] | ResNet18/256 | 400/100 | N/R | 0.872 |
| SimCLR repr. by [14] | ResNet50/2048 | 300/10 | 0.624 | 0.866 |
| Our SimCLR (2 views, ours) | ResNet18-512/128 | 128/90 | 0.679 [0.667, 0.690] | 0.888 [0.884, 0.893] |
| **Our Poly-Window (8 views)** | ResNet18-512/128 | **32**/90 | **0.680** [0.672, 0.688] | **0.891** [0.889, 0.893] |

## C. Comparison to previous work

Table II presents a direct comparison of our poly-window contrastive learning framework with representative self-supervised methods for multi-label superclass classification on PTB-XL. For comparability, all listed methods utilize ResNet-based architectures and adhere to the same pre-training and evaluation protocol.

TSSL [13] employs a ResNet18 backbone with a 256-dimensional embedding, trained for 400 epochs of pre-training and 100 epochs of linear evaluation. Among other recent contrastive methods, CLOCS [15] achieves an AUROC of 0.784, while 3KG [16] applies graph-based contrastive learning and reports an AUROC of 0.751.

Our SimCLR (2 views, ResNet18-512/128) baseline uses a slightly smaller embedding and is trained for 128 pre-training epochs, with up to 90 epochs of linear evaluation (noting that linear evaluation is typically stopped early based on validation performance, but we report results at 90 epochs for consistency). This baseline achieves an AUROC of 0.888 [0.884, 0.893] and F1 of 0.679 [0.667, 0.690], already exceeding previous results.

Our poly-window (8 views) approach further improves performance, reaching an AUROC of 0.891 [0.889, 0.893] and F1 of 0.680 [0.672, 0.688] with only 32 pre-training epochs and up to 90 epochs of linear evaluation (again, with possible early stopping). This result highlights both a state-of-the-art performance and a significant reduction in required pre-training epochs compared to previous works.

For reference, the implemented SimCLR baseline [14] uses a larger ResNet50 backbone (2048-dimensional embedding) and 300 pre-training epochs achieves AUROC 0.866 and F1 0.624. Both values are below our results. Additionally, this work [14] reported metrics by pre-training on larger, multi-institutional ECG datasets (e.g., Ningbo [23], CODE-15 [24], Chapman [25]). To focus on the method itself, our primary comparison restricts both pre-training and evaluation to PTB-XL.

In summary, with comparable ResNet architectures and consistent single-dataset evaluation, our poly-window approach achieves the highest AUROC (0.891 vs. 0.872 for TSSL [13] and 0.866 for SimCLR) and F1 (0.680 vs. 0.679 for our SimCLR baseline and 0.624 [14]) with significantly fewer pre-training epochs as the number of views increases.

TABLE III: **Wall clock time (minutes) for varying numbers of views and training epochs.** The values are shown as the mean with a confidence interval (CI) of 95% over five seeds.

| Windows | 16 epochs | 32 epochs | 64 epochs | 128 epochs | 256 epochs |
|---|---|---|---|---|---|
| 2 | $0.7 \pm 0.0$ | $1.4 \pm 0.0$ | $2.8 \pm 0.0$ | $5.4 \pm 0.1$ | $9.3 \pm 0.4$ |
| 4 | $1.1 \pm 0.0$ | $2.1 \pm 0.0$ | $4.0 \pm 0.1$ | $7.3 \pm 0.3$ | $13.8 \pm 0.6$ |
| 6 | $1.6 \pm 0.0$ | $3.2 \pm 0.0$ | $6.4 \pm 0.1$ | $12.5 \pm 0.3$ | $23.4 \pm 0.8$ |
| 8 | $2.4 \pm 0.0$ | $4.6 \pm 0.1$ | $9.3 \pm 0.2$ | $17.2 \pm 0.7$ | $32.7 \pm 1.5$ |

## D. Computational efficiency analysis

The results in Table III and Figure 2f illustrate the trade-off in poly-window contrastive learning. Increasing the number of views per ECG increases the wall clock time required for each epoch. However, the total wall clock time required to achieve optimal performance is actually lower with more views, despite the increased per-epoch cost. Specifically, in Table I, our experiments show that the 8-window model reaches its optimal validation AUROC (0.891) after only 32 epochs, whereas the 2-window baseline requires 128 epochs to get its best AUROC (0.888). The training with two windows for 128 epochs takes approximately 5.4 minutes (see Table III), while the 8-window configuration requires just 4.6 minutes for 32 epochs, which is an improvement of about 14.8% in total wall clock training time. Furthermore, the 8-view configuration (see Figure 2) not only converges faster, but also consistently yields the highest overall model performance, outperforming all other settings in terms of AUROC.

Crucially, this time savings is even more significant when considering reduced validation frequency, fewer checkpoints, and much faster hyperparameter optimization cycles enabled by rapid convergence. In our experiments, we used a single NVIDIA A100 GPU (80 GB). For both the two-view SimCLR baseline and our $M = 8$ poly-window method, we observed peak memory usage of 12.35 GB during training with a batch size of 1024 and crop size of 64. Hence the effective memory footprint is comparable for both methods under equivalent settings. For the two-view SimCLR configuration at a smaller batch size of 256, memory usage dropped to 1.2 GB.

## IV. CONCLUSIONS

In this work, we introduced a poly-window contrastive learning framework for self-supervised ECG representation learning, leveraging multiple temporally diverse views from

each recording. On PTB-XL, our approach consistently outperformed conventional two-view contrastive methods for multi-label superclass classification, achieving higher AUROC (0.891 vs. 0.888) and F1 score (0.680 vs. 0.679). Notably, poly-window learning enabled robust performance using up to four times fewer pre-training epochs (32 vs. 128) and a 14.8% reduction in total wall clock training time (4.6 vs. 5.4 minutes). Our ablation studies identified crop size and window multiplicity as the most critical hyperparameters. Furthermore, our framework outperformed recent contrastive baselines with the ResNet18 architecture and allows for fewer training epochs. The method itself is model-agnostic and could be readily applied to other deep learning architectures and biomedical time series.

However, several limitations remain. All training and evaluation in this study were restricted to the PTB-XL dataset, and generalizability to other ECG cohorts or to real-world data with broader population and device variability is untested. Our framework was evaluated exclusively on 12-lead ECG signals and only for multi-label superclass classification. Its effectiveness for single-lead ECG, other physiologic signals, or additional downstream tasks remains unexplored.

## Acknowledgment

A.F., Y.Y., and J.V.D. were supported by the Nell Hodgson Woodruff School of Nursing at Emory University. S.P. was supported by NIH R01DA040487 and in part by NSF 2112455, and NIH 2R01EB006841. In addition, R.X. and A.F. were partially supported by Georgia Clinical & Translational Science Alliance (CTSA) BERD Grant from the National Center for Advancing Translational Sciences of the National Institutes of Health under Award Number UL1TR002378. The content is solely the responsibility of the authors and does not necessarily represent the official views of the National Institutes of Health.

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
