# OpenReview forum: "Learning ECG Representations via Poly-Window Contrastive Learning"
_IEEE.org/EMBS/BHI/2025/Conference — BHI 2025_

### Official Review · Reviewer_ppMv · 2025-06-27
**Learning ECG Representations via Poly-Window Contrastive Learning**

**Confidence:** 4
**Clarity Of Writing:** good
**Clinical Significance:** good
**Methodological Novelty:** good
**Overall Rating:** 7

**Experiments And Results:**

great

**Questions For The Authors:**

Could the authors explicitly list the superclass diagnostic labels used for multi-label classification that will improve the clinical importance of the work?

What is the physiological or theoretical motivation for choosing geometric vs. arithmetic mean when aggregating contrastive loss over positive pairs?

Could the authors provide qualitative evidence showing that the learned representations capture slow-changing physiological features?

**Strengths:**

The method adapts existing ideas from contrastive learning to the ECG domain in a computationally efficient and physiologically motivated way. The proposed method significantly reduces training time and pre-training epochs, which is valuable for clinical deployment. Ablation studies are comprehensive and statistically rigorous (e.g., reporting η² effect sizes and p-values), enhancing the credibility of the conclusions. The paper includes fair and transparent comparisons against SimCLR and TSSL under consistent data and model conditions.

**Summary Of The Paper:**

This paper introduces a poly-window contrastive learning framework for self-supervised representation learning from ECG signals. The method samples multiple temporal windows from each ECG trace and maximizes agreement across all intra-instance positive pairs. The design draws inspiration from slow feature analysis to ensure temporal invariance in learned features. The authors validate their approach on the PTB-XL dataset through multi-label superclass classification, reporting performance improvements in ROC-AUC (0.891) and F1 score (0.680), alongside a reduction in training epochs (32 vs. 128) and total wall-clock time (14.8% reduction).

**Weaknesses:**

The paper mentions "multi-label superclass classification" as the downstream evaluation task but does not explicitly list the clinically meaningful superclasses used in the PTB-XL dataset. As a result, a reader who is not familiar with the PTB-XL dataset may not understand what the model is actually predicting, or how those predictions relate to clinical outcomes.

While both geometric and arithmetic means are tested, a theoretical justification or rationale for choosing one over the other (besides marginal performance gains) is missing.

Minor comments:
Please ensure that all acronyms are spelled out in full when first used in the manuscript.
Please be consistent with the use of AUROC or ROC-AUC throughout the paper.
Check for grammar to improve the readability (e.g., "Which than embedded into 128-dimensional space via ...").

---

### Official Review · Reviewer_u8e4 · 2025-07-03
**Learning ECG Representations via Poly-Window Contrastive Learning**

**Confidence:** 5
**Clarity Of Writing:** good
**Clinical Significance:** good
**Methodological Novelty:** fair
**Overall Rating:** 4
**Final Rating:** 6

**Experiments And Results:**

good

**Questions For The Authors:**

- Could the authors evaluate their method on additional datasets (PhysioNet 2021: CPSC, CPSC-Extra, Georgia, Ningbo, and/or Chapman)?
- Could the authors compare their method to existing ECG pretraining methods?

**Strengths:**

- The extension of contrastive learning to multiple temporal views (poly-window) is well-motivated by physiological properties of ECGs and formalized clearly.
 - The approach achieves comparable or superior performance to strong baselines with up to 4× fewer pretraining epochs and ~15% less total wall clock time.
 - The authors conduct 7200 validation experiments to isolate and quantify the effects of hyperparameters and design choices.
 - Even the two-view SimCLR baseline surpasses prior work, establishing a credible point of comparison.
 - The paper includes detailed descriptions of datasets, architectures, evaluation setups, and metrics with statistical reporting (e.g., CI, effect size η²).

**Summary Of The Paper:**

The authors propose a novel self-supervised learning framework for electrocardiogram (ECG) representation learning, called poly-window contrastive learning. Unlike traditional contrastive methods that use two augmented views per instance, this method extracts multiple temporally diverse windows from each ECG and enforces representation similarity across all positive pairs. The method is evaluated on the PTB-XL dataset using a ResNet18-based encoder, demonstrating that poly-window learning achieves state-of-the-art performance in multi-label classification with significantly fewer pretraining epochs and reduced wall clock time. Extensive ablation studies are presented to analyze the impact of crop size, batch size, number of views, overlap, loss function, and training epochs.

**Weaknesses:**

- All experiments are restricted to PTB-XL, which, while large and public, may not reflect broader population or device variability. This limits claims about generalizability.
 -  Lack of comparisons: many previous methods have be developed to pretrained ECG models using contrastive pretraining (i.e. 3KG, CLOCS, ECG12Net, ECG-FM)

---

### Official Review · Reviewer_GM3Y · 2025-07-14
**Poly-Window Contrastive Learning for ECG Signals: Enhanced Temporal Representation and Scalability Over Traditional Approaches**

**Confidence:** 4
**Clarity Of Writing:** great
**Clinical Significance:** great
**Methodological Novelty:** great
**Overall Rating:** 5
**Final Rating:** 6

**Experiments And Results:**

great

**Questions For The Authors:**

- The technical limitation of previous contrastive learning methods (i.e. SimCLR) is unclear to me. Why can't baseline contrastive learning methods create multiple positive pairs? Is the technical limitation due to the design of the contrastive loss function or how positive pairs were created?

**Strengths:**

- The paper presents clear motivation for the proposed method and its potential impact on ECG analysis.
- The paper presents thorough description and clear visuals to explain the poly-window approach.
- The paper provides 95CI for the results to help support the significance of the findings.
- The paper presents a comprehensive ablation study on crop size, batch size etc. to validate the effectiveness of the proposed method.

**Summary Of The Paper:**

Self-supervised contrastive learning has demonstrated strong potential in analyzing electrocardiogram (ECG) signals due to its ability to leverage unlabeled data. However, prior contrastive approaches are limited to pairwise augmented views and underutilize the full temporal structures in the signals.
This paper proposes Poly-Window Contrastive Learning to capture the full temporal structure of ECG signals by leveraging multiple augmented views from different time windows.
The paper reports that its poly-window approach outperforms several existing contrastive models on PTB-XL dataset in superclass multi-label classification. It also shows that the poly-windows approach can save wall clock time than traditional contrastive learning methods, making it more scalable for larger ECG datasets.

**Weaknesses:**

- The paper never explicitly mentions which single-lead ECG was used in the experiments, except for "lead 1" in Figure 1, making it confusing for readers to understand the context of the experiments.
- Given the wide availability of public ECG datasets, the paper is encouraged to expand its validation and also consider external validation on other datasets to further support the generalizability of the proposed method.
- The paper is encouraged to discuss the limitations of the proposed method or potential areas for future work.

---

### Official Review · Reviewer_g4iL · 2025-07-17
**Technically solid paper, but lack of clinical novelty, proper baseline, and benchmark**

**Confidence:** 4
**Clarity Of Writing:** great
**Clinical Significance:** good
**Methodological Novelty:** fair
**Overall Rating:** 3
**Final Rating:** 4

**Experiments And Results:**

fair

**Questions For The Authors:**

- Memory/compute – What is peak GPU memory and per-iteration time for M = 8 vs. two-view SimCLR at equal global batch sizes?
- Why not compare against other ECG SSL method such as 3KG and CLOCS with more datasets? Showing competitiveness with these would substantially improve the paper’s impact.

**Strengths:**

- Clear formulation of a multi-window contrastive objective with both aggregation variants.
- The method is efficient– Fewer epochs and modest wall-clock savings relative to two-view SimCLR.
- Thorough ablations on hyper-parameters (crop size, windows, overlap, etc.).
- Well-written; implementation details are explicit, aiding reproducibility.

**Summary Of The Paper:**

The paper adapts multi-crop / poly-view contrastive learning to 12-lead ECGs: each 10-s signal is split into M short overlapping windows whose embeddings are drawn together with an aggregated InfoNCE loss. This study leverages 1-D ResNet-18 on PTB-XL for 16–256 pretraining and downstream task, then frozen and linearly probed on the PTB-XL superclass labels (multi-label ROC-AUC and F1). Ablations vary crop size, number of windows, batch size and loss type.
With eight windows the encoder reaches ROC-AUC ≈ 0.891 after 32 epochs, matching two-view SimCLR trained for 128 epochs. Training wall-clock time drops around 15 %.

**Weaknesses:**

- Missing strong baselines – The study mainly focuses on the comparison between SimCLR and TSSL, nevertheless, many ECG-focused or biomedical-focused SSL methods are not discussed and compared. Such as 3KG, CLOCS, multi-channel masked autoencoder (MCMA) etc.
- Single-dataset scope – All training and evaluation are confined to PTB-XL, so generalisation to other cohorts (e.g., Chapman, Ningbo) is untested.
- The method seems to be generic rather than ECG specified, so the motivation of conducting experiments on ECG only is not well explained.
- Marginal gains – The 0.003 ROC-AUC improvement over SimCLR is within typical variance and statistical significance is not reported.